# Technical Differences over the Course of the Match: An Analysis of Three Elite Teams in the UEFA Champions League

**DOI:** 10.3390/sports11020046

**Published:** 2023-02-16

**Authors:** Marco Magni, Matteo Zago, Paola Vago, Matteo Vandoni, Vittoria Carnevale Pellino, Nicola Lovecchio

**Affiliations:** 1UCFB, Buckinghamshire New University, Manchester M11 3FF, UK; 2Department of Biomedical Sciences for Health, Università degli Studi di Milano, 20133 Milan, Italy; 3E4Sport Laboratory, Politecnico di Milano, 23900 Lecco, Italy; 4Interfaculty of Education and Medicine, Catholic University of Sacred Heart, 20162 Milan, Italy; 5Laboratory of Adapted Motor Activity (LAMA), Department of Public Health, Experimental Medicine and Forensic Science, University of Pavia, 27100 Pavia, Italy; 6Department of Industrial Engineering, University of Rome Tor Vergata, 00133 Rome, Italy; 7Department of Human and Social Science, University of Bergamo, 24129 Bergamo, Italy

**Keywords:** contextual variables, football, soccer, Champions League

## Abstract

The purpose of this study was to evaluate the evolution of technical performance indicators over the course of football matches in the UEFA Champions League. Three elite football teams were the sample of the present study and were analyzed throughout four consecutive seasons within the previously mentioned competition. Data from 15 min periods were collected from Wyscout and elaborated. The effects of match location and competition stage were analyzed on nine technical indicators, including ball possession and variables related to offense and goal scoring. The effects of independent variables were assessed both independently and combined. The results showed a significant increase in the frequency of occurrence and accuracy of most of the parameters towards the end of the match. The effect of match location was generally significant with higher rates for teams playing at home. Differences were noted between the two stages of the competition with higher values in the technical indicators registered during the group stage. The existence of significant differences between the initial and final periods of football games was suggested by the results. The so-called home advantage was confirmed. Different team approaches between longer stages (e.g., group stage) and elimination games (i.e., knockout phase) were suggested by the results.

## 1. Introduction

According to the dynamic nature of open skill activities, in which athletes need to continuously adapt their behaviors to a constantly changing environment [1], football requires players to frequently adjust their performance to the unpredictable situations of play [2]. In fact, different elements need to be identified and considered in order to fulfill team success in football [3]: the technical, tactical, physiological, and mental components [4], as well as the environmental variables (e.g., match location and match status) [5], have been broadly examined in the literature [6,7,8]. Thus, in order to effectively analyze this complex and dynamic environment, most researchers attempted to investigate the relationships between key performance factors (physical, technical, tactical) and team success [9,10]. For example, some authors analyzed the physical performance describing the match activity profiles of elite players [3,11] while others defined the technical/tactical components of the game [2,12,13,14,15]. In particular, some studies observed differences between playing positions in both physical and technical activities of players [3,16], while others found a significant influence of situational (or contextual) variables, on key sport-specific movements [17] and on technical/tactical patterns of play [12]. In this context, performance analysis tools have been used to summarize the intrinsic complexity of the football environment [18], providing a finite number of descriptive data (match statistics) as an objective and unbiased record of team performance. Following the advancements of the last two decades in video analysis systems and databases (e.g., Stats Perform, London, W2 1AF, UK), the analysis of a large sample of data became easier to perform [4,19], aiding coaches and researchers to better comprehend football matches [15]. Furthermore, match statistics identified as performance indicators have often been considered crucial to monitor teams’ activities over time [5,10]: the authors agree on recognizing the technical indicators (i.e., technical effectiveness of players) as the most accurate indices of success [4,15,20] for determining the overall outcome of football matches. In particular, a critical review conducted by Mackenzie and Cushion [6] showed that most of the research in performance analysis aimed to investigate the offensive phase of the game and the technical indicators related to goal scoring: this is in agreement with the general belief in the literature suggesting that winning teams are stronger than others in the variables related to offense [10,21,22,23]. Among the variables related to goal scoring, higher numbers of shots and shots on target have been positively associated with success, while a higher effectiveness rate (i.e., the percentage of shots on target out of the number of total shots) was found to be a characteristic of winning teams in UEFA Champions League (UCL) [13], in La Liga [10], and in the 2010 World Cup [21]. Moreover, an extended range of studies recognized a positive relationship between ball possession and success [2,4,20,24], while other authors found trivial or negative influences on winning [15]. Other relevant technical indicators related to offensive actions, such as crosses [10,19] and offensive duels were also important to enrich the debate [2,3]. The inconclusive and sometimes conflicting results in the literature suggest further analyses, accounting for situational variables and different conditions [8,10]. Indeed, authors agree on recognizing a fundamental role of the contextual variables [12,20,25], as the empirical evidence highlighted the strong influence of these factors on physical and technical outcomes [7,26]. Furthermore, Yi et al. [2] argued that the nature and the relevance of the competition are strictly related to the technical and psychological attitudes of players and teams [2,15,27]. The UCL is the most important European competition for clubs [2], consisting of a group stage and a knockout phase for the highest-level teams [21,28]. Much is known in the literature on how technical indicators influence the final match outcome, but limited information is available about how these parameters change between different contexts and evolve during the course of the match. Indeed, it is intuitive and well-known to all the stakeholders that the context highly impacts the attitudes of players, as well as coaches’ requests throughout the match. The nature of the competition itself and of its moments may cause different decisions, so it would be of interest for coaches to know how technical parameters change between the different stages. Furthermore, it would be valuable for football practice to know if specific patterns of play (i.e., technical parameters) occur more at the beginning, in the middle, or at the end of the match; this notion would allow coaches to enhance their capability to plan the strategy before the match and to read the game during it. Therefore, the aim of this study was to examine and compare the changes in key technical indicators related to goal scoring and offense in different contexts (i.e., group stage and knockout phase of the UCL, playing home and playing away) and throughout the match. Three elite-level teams were considered for this study.

## 2. Materials and Methods

### 2.1. Sample

Technical-related data from a total of 128 matches played by three clubs (Atletico Madrid, Barcelona, and Real Madrid) during four consecutive seasons (2015–2016, 2016–2017, 2017–2018, and 2018–2019) in the UCL were downloaded from the online platform Wyscout (Wyscout Spa, Chiavari, Italy) [22]. This platform provides a complete database, including the most advanced metrics used in professional performance analysis. For the purpose of this study, 56 games played in the group stage and 72 in the knockout phase were analyzed.

The four seasons were considered together, as is a common procedure in the literature [4,5,18,29]. The choice of the three teams to analyze was based on the need to evaluate teams of a similar level, in order to allow comparisons and generalization of the results: the teams selected were constantly among the four best-rated in the UEFA club coefficients within the selected seasons [30], obtaining stable results and being solidly among the best European clubs.

### 2.2. Procedures

To investigate the evolution (frequency and accuracy) of technical indicators across the selected football games, measures were compared across 15 min periods [21,29]. A 15 min interval was defined as a“segment” as displayed in Wyscout: segment 1: 1–15 min; segment 2: 16–30 min; segment 3: 31–45+ min; segment 4: 46–60 min; segment 5: 61–75 min; segment 6: 76–90+ min. Additional time was included in the last 15 min period of play. Extra times were excluded because not considered in the Wyscout database.

Two independent factors were defined: competition stage (group stage and knockout phase) and match location (matches played at “home” or “away”). Regarding the dependent variables, among the technical performance indicators related to offense, some of the most investigated variables in the professional literature, consistently reported in Wyscout, were chosen; in accordance with existing literature [4,10,14], parameters were classified into three different groups: (i) four variables related to goal scoring, (ii) nall Possession, and (iii) four variables related to offensive play, as detailed in Table 1.

### 2.3. Statistical Analysis

Gathered data from the Wyscout platform from every team and season were exported to a single spreadsheet in IBM SPSS (IBM Corp. Released 2019. IBM SPSS Statistics for Windows, Version 26.0, Armonk, NY, 10504, USA) for the analysis. The statistical analysis was conducted by means of a multivariate analysis of variance (MANOVA) [31], in order to determine which dependent variables revealed differences in terms of the following factors: home/away, group vs. knockout, and segments of the match. The level of significance was set at *p* < 0.05, and the Wilks’ lambda test was used. The statistically significant effects of different stages of the competition and different match locations were analyzed using the “estimated marginal means” (EMM), while those of segments were further analyzed by applying the Bonferroni post hoc correction [24]. In the latter, the analysis was conducted until the second level of interaction.

## 3. Results

A total of 128 games were analyzed, in which 248 goals were scored; 37.9% of goals were scored in the second phase of the UCL and, overall, 60% were scored in the second half of the match.

### 3.1. Variables Related to Goal Scoring

When playing at home (Figure 1a,b), overall the three teams showed a significantly higher number of Goals (F(1,74) = 18.722, *p* < 0.001), Shots (F(1,74) = 28.111, *p* < 0.001) and Shots on target (F(1,74) = 25.418, *p* < 0.001) than when playing away. The estimated marginal mean is generally higher or equal for these variables when playing home in opposition to when playing away (see Figure 1a,b). Considering the matches of the group stage, the three teams showed more frequent shots (F(1,74) = 10.680, *p* = 0.001) and shots on target (F(1,74) = 8.733, *p* = 0.003), than in the contest of knockout phase. This is true in particular concerning the home teams, showing higher parameters in the first phase of the UEFA Champions League. For example, the EMM of home teams for shots is higher than 3.0 per match for four out of the six segments in the group stage, while it is lower than this benchmark for every segment except the last one in the knockout phase. The same is visible for shots on target, with five out of six segments outscoring the benchmark (i.e., 1.0) in the group stage, while just two segments (i.e., 2 and 6) did it in the knockout phase (Figure 1a,b). Regarding segments, the first 15 min period showed a significantly lower value if compared to the last period for goals, shots, and shots on target, while a lower value was also shown in relation to the second period for goals and to the third and fifth segments for shots. In particular, the shots on target performed by home teams during the last segment of the matches played in the group stage were about double this parameter registered in the first segment of the same competition.

During segment 6, teams scored statistically more than in segment 3 and shot more than in segment 2, while in the fourth segment, fewer shots and shots on target were performed than in the last 15 min. In general, an increasing trend for goals, shots, and shots on target over the course of the match was visible during the group stage when teams played at home, although these results were not confirmed by the statistical analysis. It is largely confirmed that the last segment showed higher frequency for most of the goal-scoring variables if compared to the first and the second segments (e.g., the difference of more than 0.2 in EMM for home teams regarding goals, of more than 0.5 in EMM for home teams in the shots on target, Figure 1a,b). The effectiveness did not significantly change in relation to the considered factors, despite the generally higher values of the last segment when compared with segments 1, 3, and 4.

### 3.2. Ball Possession

Ball possession (Figure 2) did not change significantly in relation to match location and it remained constant during the match. Indeed, the parameters shown by this variable are very similar when comparing teams playing home and teams playing away (e.g., the 3rd and the 6th segment display about the same value between home and away teams, Figure 2). In contrast, the stage of the competition significantly impacted this variable, considering the statistically higher percentage of ball possession in the group stage (EMM = 58.3%) than in the knockout phase (EMM = 52.9%).

### 3.3. Variables Related to Offensive Play

When playing at home, teams displayed more crosses (F(1,74) = 9.350, *p* = 0.002) and accurate crosses (F(1,74) = 4.373, *p* = 0.037) than when playing away (Figure 3a). This trend was shown for most of the segments, even if some exceptions are visible, for example regarding accurate crosses (i.e., the last three segments during the knockout phase, Figure 3a). A statistically higher number of crosses was shown in the group stage if compared with the knockout phase, with EMMs of 2.951 and 2.512, respectively. A similar trend was visible for accurate crosses, specifically for teams playing at home, even if the statistical analysis did not reveal a significant effect. In contrast, teams playing away showed higher accuracy in the matches of the knockout phase of the UCL. Regarding the course of the match, in the last segment teams performed more crosses and accurate crosses than in all the other periods, with the exception of the third segment for crosses. This trend shows the higher accuracy in crosses over the course of the match, in particular for teams playing away, in the knockout phase of the UCL (the difference is about 0.5 for accurate crosses per match during the second phase of the competition, Figure 3a). Even though no statistical differences were found in the frequency of occurrence of offensive duels when considering different match locations and competition stages, a significantly higher number of offensive duels won occurred at home if compared with the frequency shown away (EMM = 5.62 vs. 5.19). In the last segment, a significantly higher number of both offensive duels (F(5,74) = 14.959, *p* < 0.001) and offensive duels won (F(5,74) = 9.088, *p* < 0.001) was registered in comparison to all the other segments. Regarding offensive duels, the third segment showed higher values than segments 1, 2, and 5. Even if not statistically significant, for these two variables it is possible to notice a trend to augment the frequency of the event at the end of every half and to decrease at the beginning of the two match sections (e.g., the only values to overcome the benchmark of EMM 15.0 for crosses are visible in the third and sixth segments; the same happened for accurate crosses using as a benchmark, EMM: 6.0). No statistically significant interactions were found between the segment and competition stage, segment and match location, and match location and competition stage.

## 4. Discussion

The current investigation studied the impact on technical indicators of match running time, match location, and different competition stages: these were, in our case, the two stages (group stage and knockout phase) of the most important tournament in Europe, the UEFA Champions League, played by three top-level teams.

### 4.1. Segments

In general, the frequency of occurrence of the technical variables considered in our study increased towards the final minutes of matches. In particular, the last segment showed, with some exceptions, more goals, shots, shots on target, crosses, and accurate crosses when compared to the previous ones. These findings reinforce the theoretical general belief of an increasing urge to play toward the end of a match [29]. As the first 15 min period displayed, generally, lower goals, shots, shots on target, and offensive duels than others, it might be suggested that at the beginning of the match teams playing in UCL usually adopt a more cautious approach. Moreover, at the beginning of the match players are in their best physical condition [9] and this may lead to higher defensive quality and organization [21].

### 4.2. Home/Away

Our results confirmed the popular vision of the advantage of playing at home in UCL [14,18], which was crucial in domestic leagues [5,10]: higher values were displayed in home matches if compared with away games in the variables related to goal scoring (goals, shots and shots on target; Figure 1) and to offensive play (crosses and accurate crosses; Figure 3). Sarmento et al. [23], analyzing 68 games from different domestic leagues (Italian Serie A, Spanish La Liga, German Bundesliga, English Premier League) and UCL found similar results, confirming the higher likelihood of teams playing at home to perform a higher number of shots, shots on target and crosses than teams playing away [23]. Effectiveness was not statistically affected by match location and competition stage. Considering the quality level of the selected teams, these data confirmed the general thought in the literature suggesting that the accuracy rates of elite players do not change depending on situational variables [12,21,25]. The trivial effect of match location on ball possession is an unexpected result, as previous findings in the literature suggested a significantly higher ball possession for teams playing at home [25,26]. This might be linked to the specific playing style of the selected teams [32] remaining consistent in different conditions.

### 4.3. Group Stage/Knockout Phase

Regarding the differences between the two phases of the UCL, interesting considerations emerged. The percentage of ball possession as well as the number of crosses, shots, and shots on target was higher in the UCL group stage than in the knockout phase. This might be related to the nature of the competition, as teams are deemed to be more cautious when facing elimination games [15]. Additionally, the quality of the opposition is often higher in the knockout phase, being the final stage of the competition. This leads to a more balanced “rapport of strength” between the clubs competing, with teams tending to display less individual and collecting behaviors than when facing lower-quality sides. Another reason might be the deliberate choice of different offensive tactics. Indeed, as the three teams were in the first four positions of the UEFA ranking in the selected seasons, it is likely that these high-performance teams showed a lower urge of attacking during the second leg of the elimination games. This is confirmed by the increase in crosses and accuracy for teams playing away in the knockout phase, showing a higher use of this tactic as time passes. Accordingly, Alves et al. [15] found lower ball possession to be a characteristic of winning teams in the final stages of international competitions. Additionally, authors found losing teams to retain more ball possession [12,25]. Therefore, it might be plausible that these teams retained less ball possession because they were winning. Further studies might implement these findings taking into account the evolving scoreline, considering the aggregate score of knockout games [33].

### 4.4. Practical Implications

The effect of match location showed significant differences for several variables, confirming the importance of what is generally referred to as “home advantage”, which should be carefully considered by coaches when preparing for an away game. The coaches’ strategy planned before the game might be more conservative in away games, knowing it is harder to win. By contrast, other coaches might prefer to try and mitigate the disadvantage by the use of a more proactive and courageous approach, trying to surprise the opposing team from the beginning of the match. Indeed, our research showed that the first 15 min seem to be the most “quiet” segment for all the variables related to goal scoring, particularly if matches are played away. Therefore, the home team might foresee a conservative approach from the away team at the beginning of the match. Additionally, the variables related to offensive play registered a higher frequency of occurrence in the sixth segment with some spurts in the third segment (crosses and offensive duels, in both two phases of the UCL). These data might reflect a common tactic to be more offensive towards the end of every half; therefore, a more aggressive pattern of play in the first minutes of every half might be an efficient offensive strategy to startle the opponents. Furthermore, more defensive attention is requested at the end of every half, as the opponents might try to apply higher pressure to the defensive line. The differences found between the group and knockout phases could represent a suggestion to intensify the approach since the first matches of UCL. This effect might be mitigated by the lower quality of opposition in the first stage of the UCL. Data on accurate crosses and offensive duels in the knockout phase showed a higher frequency for these events displayed by teams playing away in the last segment (even if these data were not confirmed by the statistical evidence). This might reflect the tendency for teams playing away during the knockout phase to boost their offensive approach, trying to force playing from side areas. Similar findings were found by Dellal et al. [16] for offensive duels and offensive duels won in the French First League, suggesting the increased use of side areas when time scores.

### 4.5. Limitations and Suggestions for Further Studies

These findings could not be immediately generalized as only three elite-level teams were analyzed: further studies should try to compare teams of different levels and leagues. Another limitation is related to the quality of opposition and the evolution of the match, which were not considered in this study and are advised to be included in further research; however, it should be considered that including these variables would open a wider stratification of interactions and situations which may reduce the immediacy of the results and so their practical readability and applicability. Additionally, the duration of the periods taken into account creates a limitation, as the last segment of every half (segment 3 and segment 6) includes the added time. Therefore, these periods might, in some cases, last longer than the other four, allowing more time for technical events to occur.

Lastly, further studies should analyze long-running competitions (i.e., domestic leagues) and compare the results with those of the elimination games in order to evaluate the impact of different types and stages of competition.

## 5. Conclusions

The results of our study are here summarized:The significant increase in the frequency of occurrence and accuracy of key parameters towards the end of the match suggested evidence of significant differences in playing practice between the initial and final periods.The effect of playing in different stages of the UCL competition highlighted the difference between the group stage (more similar to a long-running competition in its nature) and elimination rounds. Further studies should focus more on these discrepancies, evaluating the different levels of competitions and practical implications.The effect of match location was significant for most of the variables, confirming the importance of what is generally referred to as “home advantage”.Practical implications of the research referred to the knowledge of the effect of match running time on match technical performance, aiding coaches and sports scientists in their daily work while preparing athletes for these occurrences.

## Figures and Tables

**Figure 1 sports-11-00046-f001:**
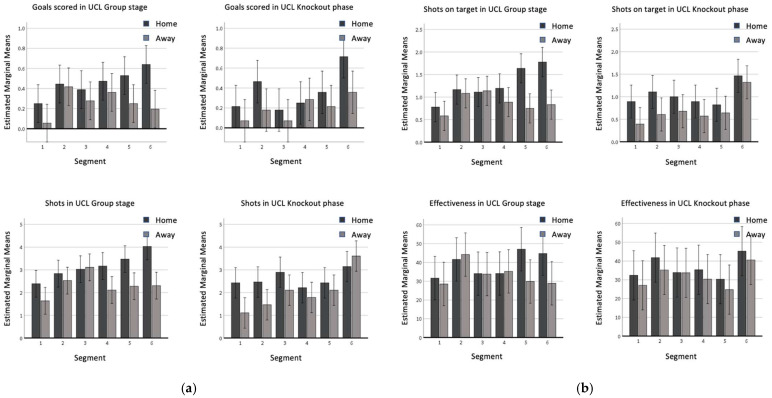
(**a**) Goals and shots in the UCL group stage and in UCL knockout phase; (**b**) shots on target and effectiveness in the UCL group stage and in the UCL knockout phase.

**Figure 2 sports-11-00046-f002:**
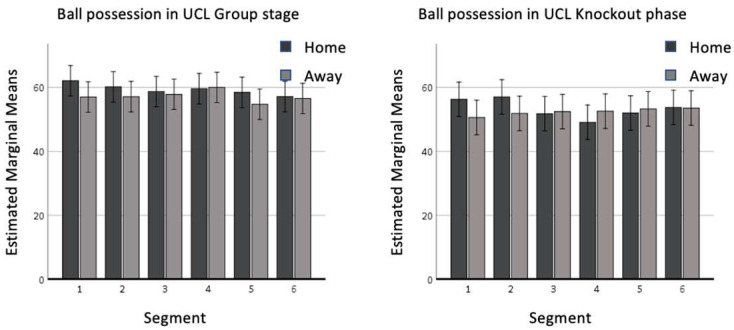
Ball possession in the UCL group stage and in the UCL knockout phase.

**Figure 3 sports-11-00046-f003:**
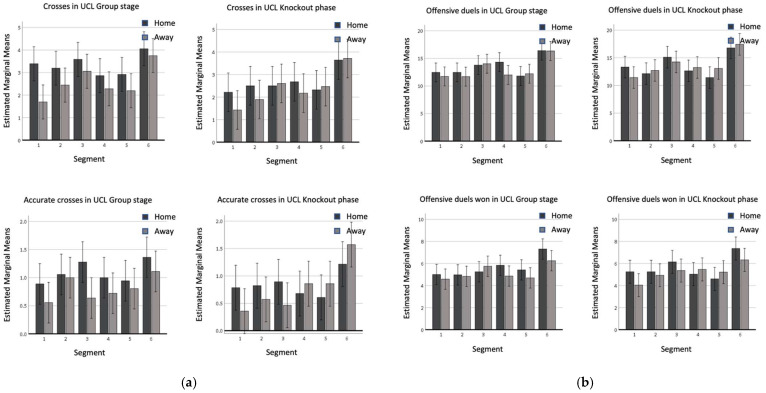
(**a**) Crosses and accurate crosses in the UCL group stage and in the UCL knockout phase; (**b**) offensive duels and offensive duels won in the UCL group stage and in the UCL knockout phase.

**Table 1 sports-11-00046-t001:** List of considered dependent variables.

Groups	Event	Operational Definition	Unit
Variables related to goal scoring	Goals	The ball crosses the goal line and is confirmed by the referee	N
	Shots	An attempt to score a goal, with any part of the body	N
	Shots on target	Any attempt to score a goal that required intervention to prevent the shot to cross the goal line	N
	Effectiveness	Shots on target × 100/shots	%
Ball possession	Ball possession	The time when a team takes over the ball from the opposing team without any clear interruption, as a proportion of the total time when the ball was in play	%
Variables related to offensive play	Crosses	Any ball sent to a teammate into the opposition team’s area from a wide position	N
	Accurate crosses	Successful crosses out of the total number of crosses	N
	Offensive duels	An attempt from a player to beat an opponent when in possession of the ball	N
	Offensive duels won	A successful attempt to beat an opponent in possession of the ball	N
Duration	Duration	The duration in minutes of the period considered	N

## Data Availability

Technical-related data were downloaded from the online platform Wyscout (Wyscout Spa, Chiavari, Italy), available at: https://wyscout.com (accessed on 21 July 2020).

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
