# Peer review of "Technical Differences over the Course of the Match: An Analysis of Three Elite Teams in the UEFA Champions League"

_sports, 2023, doi:10.3390/sports11020046_

Round 1

Author Response

dear reviewer 

please find the attached file

Reviewer 2 Report

Reviewer's opinion:

 The topic is very general - if we are talking about the observation of offensive activities - such activities should be marked in the subject of the article.

1.       In the methodological activities (observations), the criteria are given in very general terms, e.g.:

• efficiency - it's a goal, not a shot into the goal

• pass (cross) - specify, for example, an effective pass to a partner, not only into the penalty area

2. It was not stated why such observation parameters were chosen - although they are significant for offensive operations - it was necessary to argue, e.g. that professional literature indicates so, etc.

3. Observed parameters are given in very general terms, e.g. individual duels - in scientific research, praxeological indicators should be taken into account in order to fully diagnose, e.g. e.t.c.

4. The methodology does not provide a measurement tool, e.g. an observation sheet - the method of obtaining data

5. The research results lack numerical data - the presentation only in the charts is not objective in my opinion

6. Conclusions are descriptive and very general, fitting more into an overall summary. According to the reviewer, it should be short, bulleted information resulting from the author's research and reflectiveness

7. The positive dimension is the practical implications that can be used in the organized training of football players

Author Response

(The authors gave the same response as above.)

Round 2

Reviewer 1 Report

The authors have made the changes requested by the reviewers. There are some aspects that have not been reviewed due to the design of the study. I consider the article suitable for publication in this journal. 

Author Response

thank you for your positive consensum.

we add improvements in results section.

best

Reviewer 2 Report

After re-examination, I find that the authors have made most of the suggested improvements to the article. However, my further remark concerns the measurable results (in tables). In this article, these results are presented only in graphical form. In my opinion, such a form is not objective - the chart must be supported by specific numerical values (results), resulting from statistical calculations. I leave this fact to the Editor's decision

Author Response

thank you for your comments.

we add sentences in results section to lead the reader during the graph interpretation.

best
